# Altered Cerebral Processing of Videos in Children with Motor Dysfunction Suggests Broad Embodiment of Perceptual Cognitive Functions

**DOI:** 10.3390/jpm12111841

**Published:** 2022-11-04

**Authors:** Ioannis Ntoumanis, Olga Agranovich, Anna N. Shestakova, Evgeny Blagovechtchenski, Maria Koriakina, Dzerassa Kadieva, Grigory Kopytin, Iiro P. Jääskeläinen

**Affiliations:** 1Institute for Cognitive Neuroscience, HSE University, Myasnitskaya Ulitsa, 20, 101000 Moscow, Russia; 2Federal State Budgetary Institution the Turner Scientific Research Institute for Children’s Orthopedics under the Ministry of Health of the Russian Federation, 196603 St. Petersburg, Russia; 3Brain and Mind Laboratory, Department of Neuroscience and Biomedical Engineering, School of Science, Aalto University, 02150 Espoo, Finland

**Keywords:** intersubject correlation, embodied cognition, EEG, arthrogryposis multiplex congenita, obstetric brachial plexus palsy, naturalistic stimulus

## Abstract

Embodied cognition theory suggests that motor dysfunctions affect cognition. We examined this hypothesis by inspecting whether cerebral processing of movies, featuring both goal-directed movements and content without humans, differ between children with congenital motor dysfunction and healthy controls. Electroencephalography was recorded from 23 healthy children and 23 children with limited or absent arm movement due to either arthrogryposis multiplex congenita or obstetric brachial plexus palsy. Each individual patient exhibited divergent neural responses, disclosed by significantly lower inter-subject correlation (ISC) of brain activity, during the videos compared to the healthy children. We failed to observe associations between this finding and the motor-related content of the various video scenes, suggesting that differences between the patients and controls reflect modulation of perceptual-cognitive processing of videos by upper-limb motor dysfunctions not limited to the watching-mirroring of motor actions. Thus, perceptual-cognitive processes in the brain seem to be more robustly embodied than has previously been thought.

## 1. Introduction

Imagine the variety of ways in which your life experiences would have been different had you been born without an arm. Playing with other children, climbing trees, getting dressed—with a little bit of reflection, it is easy to realize that the list of activities that would have been altered grows long. According to theories of embodied cognition, our motor capabilities can shape the way we perceive and think, thus shaping our experiences [1,2,3,4]. Whether and to what extent perceptual and cognitive functions are embodied is currently one of the most heated debates in cognitive sciences and psychology [4,5,6,7,8,9,10]. The concept of embodiment has been both criticized harshly as having no scientific value [9] and defended as crucial for our understanding of human cognition [4], with some suggesting a middle ground between the embodied and disembodied cognition hypotheses [11]. Embodiment simulation is yet another term that has been introduced to describe the mechanism by means of which the interplay between brain and body emerges [12,13,14,15]. Here, we set forth to investigate this fundamental question by asking whether impairment of an aspect of one’s body affects one’s relevant perceptual-cognitive processes when watching video clips, which are increasingly used as naturalistic stimuli in neuroimaging studies [16,17,18].

An example of a similar approach, yet confined to specific functions, includes findings showing that impairing an individual’s ability to use facial muscles that support expression of particular emotions has been found to diminish the ability to recognize the relevant emotions in others [19,20]. Restricting the movements of the face has also been found to influence the perceived valence of auditory stimuli [21]. Furthermore, Parkinson’s disease, which causes impaired motor skills, has been found to weaken the processing of graspable object images [22]. On a similar research line, Beilock [23] argues that the ability to perceive the actions of others is modulated by the variations in individuals’ motor skill repertoires. If this is true, then patients with a motor impairment might be processing the movements and goal-directed behaviors of others differently to healthy controls. What remains to be further tested is whether perceptual-cognitive functions are impacted by motor impairments beyond the modality of perceiving others’ actions.

Obstetric brachial plexus palsy (OBPL) is a closed traction injury of the brachial plexus incurred during birth, with an incidence rate of 0.5 to 2.6 per 1000 live births [24]. Although the prognosis is generally considered to be good, 20–30% of patients have a residual deficit [25]. Severe OBPL can result in the permanent impairment of arm function, skeletal malformation, cosmetic deformity, behavioral problems, and socio-economic limitations [26,27,28]. Patients with obstetric palsy have defective motor programming [29] and the concept of impaired central motor programs in OBPL is also supported by observations of OBPL infants ‘‘forgetting their arm’’ during automatic movements [30]. Differences in automatic movements between the affected and unaffected side are caused by incomplete central program development and may contribute to incomplete arm function recovery following OBPL [30].

Arthrogryposis multiplex congenita (AMC) is a term used to describe a group of congenital conditions that are characterized by joint contractures in two or more body areas [31]. The symptoms of AMC are present from birth (congenital). While the precise cause may be unknown for some individuals, causes are variable and may include genetic, parental, and environmental factors, as well as abnormalities during fetal development. Individuals with AMC have limited joint movement, with or without muscle weakness, in the involved body areas. Contractures vary in distribution and severity, do not progress to previously unaffected joints, but may change over time due to growth and treatment. Depending on the underlying diagnosis, other bodily systems such as the central nervous system (CNS), respiratory, gastrointestinal, and genitourinary systems may be affected. Since the nature of this condition is considered peripheral, the putative cognitive impairments of this population are often neglected. The treatment of AMC, as well as OBPL, starts with stretching and splinting and, where necessary, patients undergo a surgery [32,33,34,35].

Combining the clinical details of these diseases with the theory of embodied cognition, a hypothesis that can be derived is that the motor deficits in these populations impact the processing of motor-related stimuli, such as the movements of others, via mirror neuron mechanisms [36]. However, in our recent studies we have obtained findings suggesting that cognition might be more embodied than this. For example, Koriakina et al. [37] revealed that the auditory and visual memory in AMC and OBPL patients are not developed in the same way as in healthy children. Blagoveshchensky et al. [38] suggest a possible dysfunction in AMC children at a cognitive level as well, as reflected in the altered pattern of their resting state brain activity. Hence, one can tentatively hypothesize that the differences between patients and healthy controls in terms of perceptual-cognitive processes may well not be limited to the processing of motor actions, but they might be more widely and robustly embodied in nature.

In the current study, we focus on the cognitive status of AMC and OBPL, by investigating whether the motor deficits in these populations affect their processing of naturalistic movie stimuli featuring human goal-directed movements, as well as non-human-containing scenes. We selected these type of actions for our study because goal-directed behavior has been found to be supported by higher motor cortical areas, such as the primary motor cortex and the premotor cortex [39]. These areas take part in specifying movement plans, as well as in translating rules for action into the execution of motor output [39]. Naturalistic stimuli were selected over an event-related potential paradigm, because the former imitates real-world situations and is considered ecologically more valid [40]. Furthermore, a promising way to investigate whether the perceptual-cognitive processes of patients are different from those of healthy controls is to predict a patient’s brain activity from a healthy subject’s brain activity [41]. This can be achieved by an intersubject correlation (ISC) analysis, which is compatible with naturalistic stimuli. ISC is a technique which identifies neural activity that is shared across individuals [16,17,18,42]. Moreover, naturalistic stimuli have been found to be more interesting and engaging than event related paradigms [43,44], which is especially important given the young age of the subjects and patients included in this study.

We first estimated the similarity between the neural responses of patients and those of healthy children. We predicted that patients would have a lower ISC of EEG compared to healthy controls, because of their putative divergent cognitive processes, in line with the theory of embodied cognition. Further, we investigated whether the degree of severity of the patients modulated their ISC. We anticipated that the less severe a patient’s condition is, the closer their neural responses to the stimuli will be to that of healthy children. Finally, using a machine-learning algorithm, we selected scenes from the movie clips with motor related content, in order to examine whether such scenes elicit patients’ divergent brain activity. On one hand, if embodied cognition involves only specific aspects such as mirroring, then the ISC of patients would be lower only during scenes with motor-related content, but not during scenes with non-human content. On the other hand, if embodied cognition is general in its manifestations, then the ISC of patients would be lower both during scenes featuring goal-directed movements and during scenes with non-human content.

## 2. Materials and Methods

### 2.1. Participants

Twenty-three healthy subjects (11 females, ages 6–16 years, mean age = 10.39 years) and 23 patients with motor dysfunction (11 females, ages 4–15 years, mean age = 8.13 years) participated in the experiment. Healthy subjects were recruited through advertisements shared on social media. The experiments with healthy controls were conducted at the HSE’s EEG laboratory in Moscow. The under aged participants were accompanied by their parents. All patients came to the H. Turner National Medical Research Center for Children’s Orthopedics and Trauma Surgery of the Ministry of Health of the Russian Federation for treatment of deformities of upper limbs and were recruited for this experiment. Informed consent was obtained from both healthy participants and patients, or their legal guardians. The project was approved by the Institutional Review Board of Ethical Committee. Overall, the experiment was carried out in accordance with the recommendations of the Declaration of Helsinki and its amendments, and the protocol was approved by the ethics committee of the National Research University, Higher School of Economics.

### 2.2. Clinical Assessment of Patients

Prior to the experiment, the motor development of the patients was assessed by neurologists. The outcome of the assessment was the patient’s diagnosis (AMC or OBPL), the estimation of the patient’s Paresis, general motor development (GMD), perinatal central nervous system damage (CNS damage) and delayed motor development (DMD). The GMD was a continuous variable, with possible values from 1 to 14, reflecting the overall motor development of a patient. A healthy child would score 14 in this scale, and the lower the score the more severe the condition. Appendix A summarizes the Paresis, CNS damage and DMD scales.

### 2.3. Cognitive Assessment of Patients

As well as the patients’ motor development, their cognitive development was also assessed prior to the experiment, following the same procedure used in Koriakina et al. [36]. In particular, we assessed their attention span, auditory memory, visual memory, and the verbally logical aspect of thinking, before the experiment. Specifically, the Digits Forward and Digits Backward tests were used to assess patients’ auditory memory and attention [45]. In order to assess patients’ visual memory, ten pictures were presented to the patients in a consecutive way (one image per second), after which they were asked to name the objects they remember [46]. In order to assess patients’ formation of generalization, conceptual development and their ability to highlight an essential feature, we used the “Sequential pictures” task, which demonstrates randomly arranged plot pictures [46]. The patient is asked to lay out the pictures in order, making up a logical story. We selected the above battery of neuropsychological assessments based on time constraints (60 min per child), the Russian nationality of our sample and its age characteristics (4–15 years). All patients’ scores in the Clinical and Cognitive assessment can be found in Table 1, along with their demographics.

### 2.4. Stimuli

Each subject was presented with 83 silent clips integrated into 4 video blocks of 6 m each. Due to technical reasons, some participants did not watch all of the videos (Appendix A). The order of the blocks was randomized across subjects, whereas the order of the clips within each block was fixed. The mean duration of the clips was 16 s and there was no narrative structure across them. Each video included scenes with human motor activity (particularly arm and leg movements), such as a child playing with toys, or engaged in sports, as well as neutral scenes without human content (Appendix A). The EEG recordings of the last two minutes of each video were omitted, as not all participants watched all the videos to the end. Notably, the videos were presented in a silent mode, because experimental subjects (of both groups) were simultaneously presented with a non-attended auditory oddball stimuli, the results of which will be separately reported elsewhere. Performing distracting tasks while watching videos has been found to diminish neural synchronization of subjects [47]. However, given the structure of the stimuli used in the current design (no narrative sequence of events, no sound), the motor-related visual information could be retrieved by the subjects, even under the potential distraction caused by the oddball task.

### 2.5. EEG Data Collection and Preprocessing

EEG activity was recorded by means of 32 electrodes for healthy children and 19 electrodes for patients, at a sampling frequency of 500 Hz. In order to not bias the Correlated Components Analysis (CorrCA), we only analyzed the recordings of the 19 electrodes that the two groups had in common. The EEG preprocessing pipeline followed previous studies [17,48]. First, the EEG segments corresponding to the duration of each video block were extracted and temporally aligned across subjects and patients. Then, the signals were high-pass filtered at 1 Hz and low-pass filtered at 50 Hz. Next, the channels whose average power exceeded the mean channel power by 4 SDs were identified and replaced with zero valued samples, so that these channels do not affect the calculation of the covariance matrices. Eye-movement related artifacts were removed by Independent Component Analysis, using the fastICA algorithm [49]. Outliers were replaced with zero, as well as the samples 40 ms around them (before and after). As outliers we classified the samples whose magnitude exceeded 3 SDs of the mean magnitude of their corresponding channel. Lastly, the time course of each channel and each subject was z-scored. Provided that the two groups of subjects were recorded with different EEG systems, we z-scored the time courses in order to control for any between-groups confounding factors. This step is typical in fMRI ISC studies [40].

### 2.6. Correlated Components Analysis

The goal of CorrCA is to find linear combinations of electrodes that are consistent across subjects and maximally correlated between them. Each such linear combination projects the data from an (N subjects × D electrodes × T time points) space to an (N × T) space, where the intersubject correlation is calculated. Parra et al. [50] offer a detailed description of the method. In brief, $R_{b}$ is the between-subjects covariance and $R_{w}$ is the within-subjects covariance. Then, the correlated components of the data are estimated so that the largest correlation between subjects is captured. It is proven that the component projections that achieve this are the eigenvectors of the matrix $R_{w}^{-1}\cdot R_{b}$ with the strongest eigenvalues [50]. The ratio of the between-subjects covariance and the within-subjects covariance of the projected data is considered the ISC.

First, the data from patients and healthy children were concatenated for each video. Based on the concatenated data, between-subject and within-subject covariance matrices were computed for each stimulus. These matrices were then averaged over the four video blocks, so that all of the stimuli correspond to the same projection vectors [48]. This allowed us to meaningfully average the ISC values of each individual across videos. Using both groups to optimize the correlated components ensured that this optimization was not biased in favor of any group [48]. After the optimization of the correlated components, we calculated the ISC of each subject in a leave-one-out approach. Specifically, we first calculated the pairwise correlations of the subject in question and each other subject in the healthy group. These pairwise correlations were then averaged to give a single number per component. This procedure was followed both for the healthy children and for the patients. We preferred the above approach, as is used in Iotzov et al. [41], as it directly tests the null hypothesis that patients are neurally synchronized with healthy individuals. The reported ISC values correspond to the sum of the three most correlated components, following previous studies [41,47,48,51], allowing us to measure the overall level of neural synchronization regardless of each component’s anatomical origin.

Given the richness of the naturalistic stimuli, we also computed ISC over sliding time windows, in order to assess the dynamics of ISC as a function of the motor-related content of short video scenes. To that end, the recordings were divided into 1.5 s sliding windows, with 1.2 s overlap. The ISC of each time window was then calculated based on the previously estimated projection vectors W [17]. In such an analysis, the selection of the window size can be a challenging step. Selecting narrow temporal windows comes with the cost of a compromised reliability of the ISC value [52]. On the other hand, selecting wide time windows comes with the cost of a compromised temporal resolution [53]. We deviated from previous EEG ISC studies, which used 5 s window lengths [17], because it is possible to display multiple stimulus movements, consecutively, within such a long period, making it difficult to classify each window to one movement category. On the contrary, 1.5 s is a reasonable period for achieving a reliable neural synchrony estimation, while capturing individual quick movements of the movies’ actors. Notably, we repeated the analysis with a set of alternative window sizes (200 ms, 500 ms, 800 ms, 1000 ms, 2000 ms, 5000 ms) and the reported results were robustly replicated.

### 2.7. Alpha Power Estimation

The estimation of each participant’s Alpha power during watching each video was achieved with the following steps. First, we projected the data to each of the three strongest correlated components. Second, we calculated the average Alpha band power in the frequency range 8–12 Hz. Third, we calculated the relative Alpha band power by dividing the average Alpha band power with the average broad-band power (0–500 Hz). This resulted in a single number per participant, per video and per component. Then, we averaged the relative Alpha band power across videos [48] and summed it over the three strongest correlated components [54]. Hence, each participant was assigned with a single number indicating their relative Alpha band power, which was then included in an ANCOVA in order to examine the effect of Group (healthy/patient) on ISC while controlling for Alpha power.

### 2.8. Automatic Annotation of Movements

The detection of arm and leg movements in the video stimuli was achieved in two steps. First, a machine learning algorithm was employed to detect the arms and legs in each video frame of our stimuli (OpenPose demo [55]). This algorithm is robust in detecting multiple scales and multiple people displayed on screen. Second, based on the Euclidean distance of each limb’s screen coordinates between consecutive video frames, we detected the frames containing arm or leg movement. The Euclidean distance had to be between two thresholds, so that we could identify when a movement occurred. The lower threshold served to filter out small, non-significant movement, for example, when the camera was not totally stable. The upper threshold served to filter out scene transitions. The optimal thresholds were selected by visually inspecting the output, since there is no ground truth in terms of when a movement occurred. This procedure resulted in a time series of motion indicators, which was then temporally aligned with the time-resolved ISC. That is to say, each time window was classified as «arm», «leg», «both» (both arm and leg) or «neither» (neither arm nor leg), depending on the dominant movement contained at the corresponding period. Notably, these four conditions were disjoint.

## 3. Results

### 3.1. Patients’ Neural Responses Are Dissimilar to Those of Healthy Controls during Movie Watching

First, the ISC was computed in a leave-one-out approach for each video; then, each subject’s ISC values were averaged across videos. This resulted in a single ISC value per subject, denoting how much this subject’s brain activity was correlated with all the members of the healthy group [41]. When assessed by a two-sample *t*-test, the ISC of healthy subjects was significantly higher than that of patients, for all videos (T(28.4) = 14.8, *p* < 0.0001, Cohen’s d = 4.360; Figure 1A). The spatial distribution of the three strongest components was found to be similar to those in previous studies [17,41,47,48]. The first component revealed a strong positivity at occipital sites, consistent with visual processing (Figure 1B). This suggests that the highest ISC during video watching was achieved by similar visual processing of the stimuli. Overall, the estimated correlated components were moderately perceptual and cognitive and not predominantly motor. For each subject, we report the average ISC across videos, because the videos were of similar content, with no narrative structure within them and they were themselves a series of shorter clips. In addition, all videos had a common set of correlated components, therefore, averaging is valid from a psychophysiological perspective. Repeating the analysis for each video separately robustly replicated the ISC difference between the two groups (Appendix A). Notably, these analyses revealed such a strong difference between the groups that in most cases there was no between-groups overlap in terms of ISC.

Provided that the sample of both groups consisted of children, with ages ranging from 4 to 16 years old, we also examined the potentially confounding role of age in the significant between-group differences. An ANCOVA with age as confounding variable revealed that the group differences remain statistically significant, after controlling for age (F(1,43) = 186.721, *p* < 0.0001, generalized eta-squared = 0.813; Figure 2A,B), while the interaction Age*Group was found insignificant (F(1,42) = 2.630, *p* = 0.112; generalized eta-squared = 0.059). Moreover, the correlation coefficient between age and ISC was neither statistically significant for the healthy group (r = 0.350, *p* = 0.101) nor for the patient group (r = −0.001, *p* = 0.995).

Similarly, we investigated the potentially confounding role of attentional engagement to the videos in the significant between-group ISC differences. Alpha power is a neural measure known to be modulated by attention [47,56]. An ANCOVA, with Alpha power as its confounding variable, revealed that the group differences remain statistically significant, after controlling for Alpha power (F(1,43) = 177.343, *p* < 0.0001, generalized eta-squared = 0.805; Figure 2C,D), while the interaction Alpha power*Group was found insignificant (F(1,42) = 0.158, *p* = 0.693; generalized eta-squared = 0.004). Moreover, the correlation coefficient between Alpha power and ISC failed to reach statistical significance both for the healthy group (r = −0.080, *p* = 0.717) and for the patient group (r = 0.062, *p* = 0.777).

Although patients’ neural responses to the videos proved to not be similar to that of healthy controls, they were significantly correlated among them. For this analysis, we separately estimated the optimal correlated components by including only the EEG data of patients; then, we calculated the leave-one-out ISC of patients. A one-tailed one-sample Wilcoxon Signed Rank test showed that the neural synchronization was significantly greater than zero (*p* = 0.018, effect size = 0.438; Appendix A).

### 3.2. ISC Was Not Associated with the Diagnostic Status of the Patients

After patients’ clinical assessment by neurologists, each patient’s clinical condition was characterized by numerous diagnostic measurements (Table 1). The correlational analysis revealed that none of these diagnostic measurements were associated with patients’ neural dissimilarity to healthy controls. In particular, ISC was not associated with Perinatal CNS damage (τ = −0.080, *p* = 0.638), nor with Delayed Motor Development (τ = 0.059, *p* = 0.720), nor with Paresis (τ = 0.114, *p* = 0.507). The Kendall rank correlation coefficient was preferred over Pearson’s correlation coefficient, due to the ordinal nature of the diagnostic variables CNS, DMD and Paresis [57]. Moreover, ISC was not correlated with the patient’s General Motor Development (Pearson’s correlation coefficient r = 0.013, *p* = 0.952). Notably, the above statistics were calculated after collapsing the data from the two patient groups, so that an overall association between the motor development and ISC is examined.

The same associations were investigated for each patient group separately, with null findings emerging once again (Appendix A). Notably, the range of the CNS damage, DMD and Paresis was limited within each group, especially for OBPL patients, which might have allowed a ceiling effect to emerge, driving the null findings. That is to say, in every patient, the clinical condition might have been severe enough to exert maximal effects on the perceptual-cognitive processes indexed by the ISC. Finally, the difference of ISC between AMC and OBPL failed to reach significance (*t*-test, T = −0.954, *p* = 0.352).

### 3.3. ISC Was Lower in Patients Irrespective of Video Clip Content

To test the hypothesis that embodied cognition is limited to the mirroring of others’ motor actions, the ISC was calculated in a sliding-windows fashion (1.5 s window length, 1.2 s step size) and each time window was labelled, depending on the dominant movement contained at the corresponding period. In order to accurately detect the scenes containing arm- or leg-related movements, we automatically annotated the videos using a machine learning. For each video, each window and each patient, we obtained an ISC value denoting how correlated the neural activity of the patient in question was to the neural activity of the healthy cohort, at the time point of interest. A mixed-effects linear model was developed to predict the ISC in each time window, based on the dominant movement in the time window in question, the patient’s diagnostic and cognitive scores, as well as the interaction of these scores with movement (Appendix A). None of the independent variables proved to significantly contribute to the model, suggesting that neither the motor-related content of the stimuli nor its interaction with patients’ diagnostics were related to the ISC.

A mixed-effects model was preferred over a typical fixed-effects one, because ISC has been found to significantly vary across individuals [41]. Including the subject identity and its interaction with movement stimulus as random effects allowed us to control for the intersubject variability of ISC. The model was developed in R, using the lme4 package [58].

## 4. Discussion

Whether and to what extent perceptual and cognitive functions are embodied—from no embodiment, via playing a role in the process of mirroring others’ actions, to broad embodiment of perception and cognition—is currently one of the most heated debates in cognitive sciences and psychology (for a review, see Farina [4]). Here, we examined this by empirically testing whether the impairment of an arm impacts perceptual-cognitive processes. We demonstrated that the EEG responses of AMC and OBPL patients to naturalistic video scenes were more variable than that of healthy controls. In fact, as can be seen in Figure 1, the between-group ISC differences were found to be non-overlapping, thus demonstrating modulation of perceptual-cognitive processing of naturalistic stimuli by upper-limb motor dysfunctions.

One of our primary hypotheses was that the patients would process the stimulus less similarly to the healthy controls, especially during the observation of human actions related to patients’ dysfunction (i.e., arm/leg movement), due to the neurological basis of the congenital motor dysfunction. However, the neural synchronization of the patients was similarly observed in all of the categories of motor-related video scenes, suggesting that ISC is independent to the stimulus’ motor-related content in this population. Therefore, the thesis that embodiment is rather general in its manifestations, rather than being limited to specific cognitive aspects closely related to an agent’s sensorimotor capacities, is supported [38]. In addition, our findings agree with the idea of embodied simulation [14], and with the crucial role the human body plays in the constitution of how we understand the world of others [59].

Clinical indices of the motor impairment severity of patients were not associated with the ISC differences to the healthy cohort. This implies that regardless of the severity of a patient’s condition, their neural responses are divergent from that of healthy children. This could be a ceiling effect, i.e., the effects on perceptual-cognitive functions were already caused by the least severe limitations in arm functionality. Indeed, the arm movement impairment in each of the patients was severe. Further patient cases with a wider range of diagnostic scores might help us to better understand the association between the severity of a patient’s clinical condition and their neural synchronization with healthy children.

It can be noted that in the present study, the spatial distribution of the three strongest correlated EEG components is consistent with previous studies [17,47,48,51]. However, the activation pattern at the temporal lobes found in previous studies (e.g., Dmochowski et al. [17]) did not emerge here, perhaps because in the current design the silent video clips were presented with non-attended auditory oddball stimuli, which was not temporally aligned across participants. The results of the oddball stimulus sequences will be reported, separately, elsewhere.

Intersubject correlation has previously been proposed as an estimate of attention [54]. Individuals with a low attention span are therefore likely to exhibit lower ISC during naturalistic stimuli, because they are expected to be easily distracted. However, this did not emerge in the current study, as the mixed-effects model revealed no association between attention span and ISC. Hence, patients’ low ISC scores appear to not be driven by a potential impairment of normal attention span. Furthermore, Alpha power proved to not be significantly correlated with ISC, and after adjusting for it via an ANCOVA, the patients still had significantly lower ISC than healthy controls. Hence, Alpha power, which is known to be modulated by attention [54,56], had no confounding role in our findings. Future studies estimating participants’ attention to the stimuli in different means (e.g., with eye-tracking) shall help us better understand the role of attention in the observed significant difference in ISC between the two groups.

Certain limitations of this study need to be taken into account. First, the age range of both patients and healthy children was wide (4–16 years). However, no correlation between age and ISC was found. Previous studies have found that neural variability increases with age [60], but this effect was observed after the age of 20 years. Notably, the correlation between age and ISC was very close to zero for the patient group, suggesting that the motor dysfunction modulates the perceptual-cognitive processing of naturalistic stimuli from a very early age. Second, only eight patients watched the “Video 4” stimulus. Although there is no EEG ISC study with such a low number of participants, Dmochowski et al. [51] have reported results from an analysis consisting of 12 subjects. Cohen & Parra [48] also mention that the method can be applied with such a low number of subjects. Given the consistency of the results across the four video stimuli, we argue that the findings are not biased as a result of the low or unbalanced sample size. Third, although the differences found between healthy controls and patients are strongly significant, this study does not validate ISC as a diagnostic tool, as the sample size is too low for such an implication. Fourth, prior to participating in the experiment, the patients were assessed by a psychologist, possibly making the whole procedure exhausting for them. This might have resulted in a lower level of engagement with the videos, and subsequently to lower ISC values. However, this is refuted by the lack of differences in the level of Alpha power.

Taken together, the findings of this study suggest that motor dysfunction affects not only features of cognition which are related to the processing of movements, such as mirroring, but it appears to have a general effect on the cerebral processing of naturalistic videos. This is in line with the hypothesis that cognitive and perceptual processes widely depend on our sensorimotor capacities and how we interact with the world [2,4,6,59]. Further studies could address whether and how the cognitive-perceptual processes of patients with motor dysfunction are altered by a recovery of their motor capacities, and by investigating the influence of a range of other types of motor dysfunctions on processing of naturalistic stimuli to further enhance our understanding of how perceptual cognitive functions are embodied.

## 5. Conclusions

Patients with Arthrogryposis multiplex congenita or Obstetric brachial plexus palsy exhibit divergent neural responses to naturalistic videosThe observed effect was not associated with the severity of the patients’ clinical conditionThe observed effect was not associated with the motor-related content of the stimuli, suggesting broad embodiment of perceptual cognitive functions

## Figures and Tables

**Figure 1 jpm-12-01841-f001:**
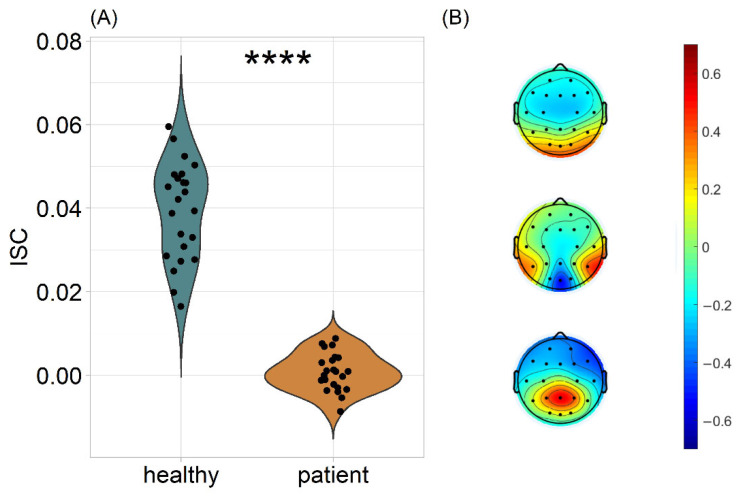
(**A**) Leave-one-out ISC as the sum of the three strongest components. Each subject corresponds to a dot representing how similar this subject’s brain activity was to the group of healthy children. The statistical comparison was conducted using a two-samples *t*-test, after establishing the normal distribution of both groups’ ISC values. * *p* < 0.05, ** *p* < 0.01, *** *p* < 0.001, **** *p* < 0.0001. (**B**) Scalp projections of the three strongest correlated components.

**Figure 2 jpm-12-01841-f002:**
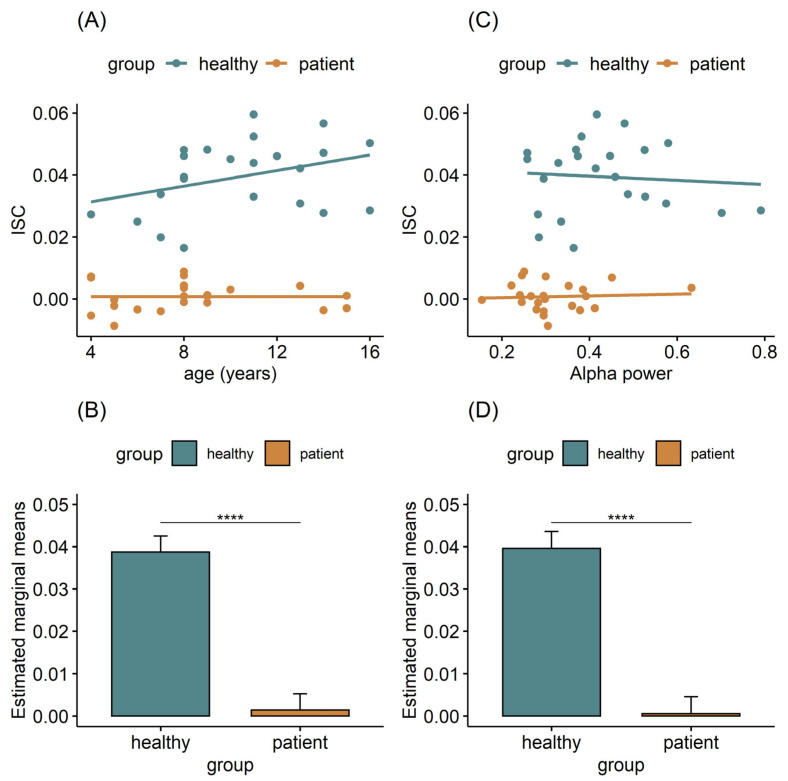
An ANCOVA was run to determine the effect of Group on ISC, after controlling for the age and the relative Alpha band power of participants. (**A**) Correlation of age and the leave-one-out ISC as the sum of the three strongest components averaged across videos. (**B**) Estimated marginal means. The *y*-axis denotes the mean ISC of each group, adjusted in terms of age. **** *p* < 0.0001, after Bonferroni adjustment. (**C**) Correlation of Alpha power and the leave-one-out ISC as the sum of the three strongest components averaged across videos. (**D**) Estimated marginal means. The *y*-axis denotes the mean ISC of each group, adjusted in terms of Alpha power. **** *p* < 0.0001, after Bonferroni adjustment.

**Table 1 jpm-12-01841-t001:** Patient Demographics and scores in the clinical and cognitive assessment.

Code	Age	Sex	Diagnosis	Paresis	CNS Damage	DMD	GMD	Attention	Auditory Memory	Visual Memory	Story
P1	8	F	OBPL	4	3	4	7	5	5	6	4
P2	6	M	AMC	3	2	3	5	1	3	4	2
P3	15	F	OBPL	4	3	3	6	3	6	3	4
P4	5	F	AMC	1	3	3	6	3	3	6	3
P5	9	F	OBPL	4	3	3	6	3	5	3	4
P6	7	M	AMC	1	2	2	6	3	4	2	4
P7	4	M	AMC	1	3	2	5	4	3	3	4
P8	8	F	AMC	1	2	1	3	2	3	3	3
P9	13	M	OBPL	4	3	4	12	5	7	10	2
P10	5	M	OBPL	4	4	4	13	3	5	6	1
P11	4	F	AMC	1	3	2	5	3	5	5	3
P12	10	F	OBPL	4	3	4	11	3	7	6	2
P13	8	F	AMC	1	2	3	5	2	4	3	4
P14	4	F	AMC	1	1	1	2	1	1	4	1
P15	8	M	AMC	1	2	2	4	2	4	5	3
P16	8	M	OBPL	4	3	4	12	3	6	6	3
P17	15	M	AMC	1	2	2	4	4	6	7	3
P18	5	M	AMC	1	2	2	4	2	3	2	1
P19	9	F	AMC	1	1	2	3	2	5	6	3
P20	14	M	OBPL	4	3	4	12	5	7	9	4
P21	8	M	AMC	1	2	1	3	2	3	5	3
P22	5	M	AMC	1	3	3	8	3	5	2	2
P23	9	F	AMC	1	3	3	6	3	7	4	5

## Data Availability

The conditions of our ethics approval do not permit public archiving of participant data, since patients were included in the current study. Code for conducting correlated component analysis has been previously published by Parra Lab (https://www.parralab.org/isc/, accessed on 15 August 2022). For the automatic annotation of movements, we implemented the OpenPose github repository (https://github.com/CMU-Perceptual-Computing-Lab/openpose, accessed on 30 May 2022). Additional code used for statistics and visualization based on the ISC analysis results, as well as any additional information required to reanalyze the data reported in this paper is available from the corresponding author upon request.

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
