# Peer review of "Altered Cerebral Processing of Videos in Children with Motor Dysfunction Suggests Broad Embodiment of Perceptual Cognitive Functions"

_jpm, 2022, doi:10.3390/jpm12111841_

Round 1
Reviewer 1 Report
The paper is interesting, it seems to be correct. Although it seems to be a bit general/introductory in the analyzes and it is a pity that the neuropsychological literature on embodiment has not been more widely engaged by authors (e.g. special issue of Cortex [Embodiment disrupted: Tapping into movement disorders through syntax and action semantics] , 100 (March, 2018). And a larger number of subjects would be desirable to study.
I have only minor comments:
[51-52] Authors should look on new critical research on the issue: Wołoszyn, K., Hohol, M., Kuniecki, M. et al. Restricting movements of lower face leaves recognition of emotional vocalizations intact but introduces a valence positivity bias. Sci Rep12,16101 (2022). https://doi.org/10.1038/s41598-022-18888-0
[108] The citation needs to be corrected.
[394-397] I would remove this sentence because in my opinion it introduces theoretical confusion. The extended mind research does not fit the research presented in the review paper.
Author Response
The paper is interesting, it seems to be correct. Although it seems to be a bit general/introductory in the analyzes and it is a pity that the neuropsychological literature on embodiment has not been more widely engaged by authors (e.g. special issue of Cortex [Embodiment disrupted: Tapping into movement disorders through syntax and action semantics] , 100 (March, 2018). And a larger number of subjects would be desirable to study.
We thank the reviewer for this reference. We browsed the articles included in Cortex's special issue on embodiment and added relevant references in our revised manuscript.
In terms of studying a larger number of subjects, we agree that it would indeed be interesting to see how our results generalize to a wider population. However, it was difficult to find more patients for our study. Notably, the final sample size provided sufficient statistical power for our data analysis.
[51-52] Authors should look on new critical research on the issue: Wołoszyn, K., Hohol, M., Kuniecki, M. et al. Restricting movements of lower face leaves recognition of emotional vocalizations intact but introduces a valence positivity bias. Sci Rep12,16101 (2022). https://doi.org/10.1038/s41598-022-18888-0
We thank the reviewer for this interesting reference. We have added this citation in our revised manuscript, as well as another one related to this topic (Ponari et al., 2012). This way, we also further engage the neuropsychological literature of embodiment in our article, which was the reviewer's first point.
[108] The citation needs to be corrected.
This citation has been corrected.
[394-397] I would remove this sentence because in my opinion it introduces theoretical confusion. The extended mind research does not fit the research presented in the review paper.
This sentence has been removed.
Reviewer 2 Report
The authors in manuscript entitled “Altered cerebral processing of videos in children with motor dysfunction suggests broad embodiment of perceptual cognitive functions” have suggested that motor dysfunctions affect cognition. The study predicted that patients will have lower intersubject correlation (ISC) of EEG as compared to healthy controls.
Strengths of the study:
- This manuscript has written in descriptive manner.
- The reference list is updated, but there is need to add some more references.
- The research article found that the clinical indices of motor impairment severity of patients were not associated with their ISC difference from the healthy cohort, which is regardless of the severity of a patient’s condition but neural responses are divergent from that of healthy children. Findings of this study suggest that motor dysfunction affects not only features of cognition which are related to the processing of movements.
There are some issues with this article, if these issues are going to resolve then the quality of the paper is suitable for publication.
1) In a part of the introduction, it should be crisp and brief about the focused study. There should be avoided to include unnecessary details.
2) Some fatigue treatment’s clinical trial related references should be included.
3) There are few typos and English grammar errors which should be rectify.
4) Discussion part should be crisp, clear and well structured.
5) Conclusion prospective must be written crisp and clear.
Author Response
The reference list is updated, but there is need to add some more references.
We thank the reviewer for this suggestion. We have now added seven more references in the revised manuscript, related to embodiment and treatment of the motor dysfunctions in question (see below).
In a part of the introduction, it should be crisp and brief about the focused study. There should be avoided to include unnecessary details.
We thank the reviewer for this suggestion. In the revised version of our article, we have tried to make the goal of our study clearer in the first and the last paragraph of the Introduction.
Some fatigue treatment’s clinical trial related references should be included.
Four references related to the treatment of AMC and OBPL have been added in the revised manuscript.
There are few typos and English grammar errors which should be rectify.
We read through the text again and we corrected any errors we noticed.
Discussion part should be crisp, clear and well structured.
We thank the reviewer for this advice. In the Discussion of the revised manuscript, we first remind to the readers the goal of our study, what were our research hypotheses and whether they were supported by the data, what are the limitations of our study and we finish with a summary of the main findings. We would be happy to restructure the Discussion if the current version is confusing.
Conclusion prospective must be written crisp and clear.
We thank the reviewer for this suggestion. We have now added a Conclusions section in our revised manuscript, where we clearly state the main findings of our study.